# Inflammasome in ALS Skeletal Muscle: *NLRP3* as a Potential Biomarker

**DOI:** 10.3390/ijms22052523

**Published:** 2021-03-03

**Authors:** Leticia Moreno-García, Francisco J. Miana-Mena, Laura Moreno-Martínez, Miriam de la Torre, Christian Lunetta, Claudia Tarlarini, Pilar Zaragoza, Ana Cristina Calvo, Rosario Osta

**Affiliations:** 1Centro de Investigación Biomédica en Red de Enfermedades Neurodegenerativas (CIBERNED), Department of Anatomy, Embryology and Animal Genetics, University of Zaragoza, Agroalimentary Institute of Aragon (IA2), Institute of Health Research of Aragon (IIS), Calle Miguel Servet 13, 50013, Zaragoza, Spain; leticiamoreno@unizar.es (L.M.-G.); jmiana@unizar.es (F.J.M.-M.); lauramm@unizar.es (L.M.-M.); mtorre@unizar.es (M.d.l.T.); pilarzar@unizar.es (P.Z.); accalvo@unizar.es (A.C.C.); 2NEMO (NEuroMuscular Omnicentre) Clinical Center, Fondazione Serena Onlus, Piazza dell’Ospedale Maggiore, 3, 20162 Milan, Italy; christian.lunetta@centrocliniconemo.it; 3Medical Genetics Unit, Department of Laboratory Medicine, Niguarda Ca’ Granda Hospital, Piazza dell’Ospedale Maggiore, 3, 20162 Milan, Italy; claudia.tarlarini@ospedaleniguarda.it

**Keywords:** neuroinflammation, NLRP3, inflammasome, biomarker, amyotrophic lateral sclerosis (ALS), skeletal muscle, SOD1G93A, blood, ALS patients

## Abstract

Since NLRP3 inflammasome plays a pivotal role in several neurodegenerative disorders, we hypothesized that levels of inflammasome components could help in diagnosis or prognosis of amyotrophic lateral sclerosis (ALS). Gene and protein expression was assayed by RT-PCR and Western blot. Spearman’s correlation coefficient was used to determine the linear correlation of transcriptional expression levels with longevity throughout disease progression in mice models. Kaplan-Meier analysis was performed to evaluate MCC950 effects (NLRP3 inhibitor) on lifespan of SOD1G93A mice. The results showed significant alterations in NLRP3 inflammasome gene and protein levels in the skeletal muscle of SOD1G93A mice. Spearman’s correlation coefficient revealed a positive association between *Nlrp3* transcriptional levels in skeletal muscle and longevity of SOD1G93A mice (r = 0.506; *p* = 0.027). Accordingly, NLRP3 inactivation with MCC950 decreased the lifespan of mice. Furthermore, *NLRP3* mRNA levels were significantly elevated in the blood of ALS patients compared to healthy controls (*p* = 0.03). In conclusion, NLRP3 could be involved in skeletal muscle pathogenesis of ALS, either through inflammasome or independently, and may play a dual role during disease progression. *NLRP3* gene expression levels could be used as a biomarker to improve diagnosis and prognosis in skeletal muscle from animal models and also to support diagnosis in clinical practice with the blood of ALS patients.

## 1. Introduction

Amyotrophic lateral sclerosis (ALS) is a neurodegenerative disease characterized by atrophy and paralysis of voluntary muscles as a result of the progressive loss of upper and lower motor neurons. Most cases of amyotrophic lateral sclerosis are sporadic (SALS), and approximately 5–10% represents the familial form of the disease (familial amyotrophic lateral sclerosis (FALS)), in which 20% have a *SOD1* gene mutation. Although SALS presents a low incidence (1.89 people per 100,000/year) and prevalence (5.2 people per 100,000), it is one of the most common neurodegenerative diseases. Unfortunately, patients have an average lifespan of less than five years after the onset of the first symptoms, which could pass unnoticed.

Current diagnostic procedures are based on a series of clinical criteria that on many occasions are not able to establish a definitive diagnosis or determine the prognosis of disease. For this reason, it is essential to improve the actual methods in order to start earlier palliative treatments and have greater control over disease progression, thus improving both lifespan and quality of life as much as possible. During the last decades, a wide variety of biomarkers have been studied in ALS involved in different molecular mechanisms that inevitably lead to motor neuron degeneration (excitotoxicity, oxidative stress, mitochondrial dysfunction, aggregation of misfolded proteins). However, today there is no sensitive and specific molecular biomarker of ALS [1,2,3].

Whereas the central nervous system (CNS) has traditionally been considered an immunologically privileged tissue, there is considerable evidence to support the presence of immune and inflammatory abnormalities in ALS. Neuroinflammation, characterized by activated microglia, astrogliosis and immune cell infiltration, is a prominent pathological finding in the spinal cord of patients with ALS (both familial and sporadic) and in murine models of the disease [4]. It has been seen that the main effector cells of neuroinflammation interact in a context- and time-dependent manner, mediating both neuroprotective and neurotoxic effects [5]. Thus, having this evidence that there is a local and systemic alteration of the immune system in ALS, it is very important to characterize it in order to determine if it is harmful or beneficial in the progression of the disease.

In the last years, it has been seen that neuroinflammation is mediated by cytosolic protein complexes, known as inflammasomes, whose function is to act as intracellular sensors to detect external signals (PAMPs, pathogen-associated molecular patterns) and internal danger signals (DAMPs, damage-associated molecular patterns). The best characterized inflammasome is NLRP3, which is composed of the NOD-like receptor pyrin domain containing protein 3, the adaptor protein ASC and pro-caspase 1. Although it has been shown that this inflammasome plays an important role in neurodegenerative disorders, such as multiple sclerosis, Alzheimer’s disease and Parkinson’s disease, little is known about its implication in ALS. Recently, several studies have shown the activation of NLRP3 inflammasome in the brains of SOD1G93A mice [6], SOD1G93A rats [7] and SALS patients [8] and also in the spinal cord of the SOD1G93A mouse model and SALS patients [9]. However, its involvement in ALS has not been studied in depth and even less in other tissues than CNS. Despite numerous studies that support the advantages and benefits of therapies targeting skeletal muscle in ALS, it is still unknown if inflammation occurs in muscle and near neuromuscular junctions, and how it could contribute to the survival of motor neurons, neuromuscular innervation or motor dysfunction in ALS [10].

In light of all this, we hypothesized that NLRP3 inflammasome activation could enhance skeletal muscle degeneration in ALS, and that therefore, its components levels could predict the disease progression. Thus, our main objective was to analyze in depth NLRP3 inflammasome expression and activation in this tissue in order to further understand muscle pathology in ALS and provide an early diagnosis and prognosis for an efficient treatment.

The findings in this study revealed that NLRP3 may be involved in ALS pathogenesis and that its gene expression levels could be used as a biomarker to improve diagnosis and prognosis in skeletal muscle from mouse models and also to support diagnosis in blood samples of ALS patients.

## 2. Results

### 2.1. NLRP3 Inflammasome Is Upregulated in Skeletal Muscle of SOD1G93A Mice

In order to investigate the activation of NLRP3 inflammasome in skeletal muscle, we first analyzed the gene and protein levels of NLRP3 inflammasome components (NLRP3, ASC, caspase 1 and IL1β) in the quadriceps of wild type (WT) and transgenic SOD1G93A mice at the asymptomatic (P50), early symptomatic (P75) and terminal stage of the disease (P120) (Figure 1).

With respect to gene levels (Figure 1A), most of the transcriptional levels of inflammasome components were significantly elevated at P50 in SOD1G93A animals compared to WT, suggesting a possible activation of NLRP3 inflammasome at earlier stages. At P75 and P120 no significant differences were observed, except for *caspase 1* levels that were significantly increased in SOD1G93A animals compared to WT at P120.

Regarding the protein expression levels, an opposed profile pattern was observed (Figure 1B). The translational levels of inflammasome components were significantly elevated at P120 in SOD1G93A animals compared to WT, while mean values tended to be increased in all the components except for NLRP3, which was observed at P75.

Interestingly, mature IL1β and active caspase 1 were detectable in SOD1G93A animals, but not in WT mice (Figure 1C). This would suggest that NLRP3 inflammasome is only activated in skeletal muscle from ALS mouse models, especially during the terminal stage, contributing thus to its pathology.

Consequently, our next step was to explore the prognostic nature of the NLRP3 inflammasome to validate its potential role during the disease progression.

### 2.2. Increased Transcriptional Levels of Nlrp3 in Skeletal Muscle at Earlier Stages Correlated with a Higher Survival Rate in SOD1G93A Mice

Skeletal muscle serial biopsies were performed on transgenic SOD1G93A mice at three disease stages: the early symptomatic (75 days), symptomatic (105 days) and endpoint stages. In this last stage, the longevity of each animal was different (ranging between 120 and 160 days), and therefore we evaluated the possible correlation between the longevity of the animals and the variation of the transcriptional levels during disease progression.

Among the four genes studied (Figure 2), the transcriptional levels of *Nlrp3* were the only ones that correlated significantly and positively with the longevity of the transgenic SOD1G93A mice (r = 0.506, *p* = 0.027) (Figure 2A). This positive correlation suggested that the animals that survived longer displayed higher transcriptional levels of this gene during disease progression, especially between the early symptomatic and symptomatic stages. Consequently, *Nlrp3* could be considered a prognostic biomarker of longevity in this animal model.

Contrary to our initial hypothesis, but in accordance with the M1/M2 paradigm in inflammation, this finding showed that NLRP3 may exhibit an unexpected regenerative capacity to compensate muscle damage induced by the neurodegenerative progression of the disease, since the transgenic SOD1G93A mice that survived longer also exhibited increased levels of this gene.

In view of these findings, our next step was to inactivate NLRP3 in this animal model to elucidate whether NLRP3 certainly could induce a beneficial role during the disease progression.

### 2.3. Inactivation of NLRP3 with MCC950 Exerted a Detrimental Effect on the Longevity of SOD1G93A Mice

In order to determine the effect of NLRP3 on the longevity of SOD1G93A mice, twenty-seven animals at age of 70 days (early symptomatic stage) were subjected to intraperitoneal injections of MCC950 or PBS (control) three times per week until death.

MCC950 is a potent and selective inhibitor of NLRP3 which closes its active conformation to an inactive state, thus preventing the assembly and activation of NLRP3 inflammasome [11,12,13]. Considering the findings obtained previously with the longevity of the mice, the inhibition of NLRP3 at the early symptomatic stage might enhance a shorter lifespan in this animal model.

MCC950-treated mice exhibited a lower mean lifespan (113.571 days ± 18.48) compared to untreated controls (128.692 days ± 7.54, log-rank test *p* = 0.025) (Figure 3), with MCC950 treatment being particularly relevant in the early stages. Interestingly, this finding is in accordance with the result obtained in our previous correlation study between longevity and the transcriptional expression levels (Figure 2), thus supporting the hypothesis that NLRP3 could exert a dual role in ALS pathogenesis with a compensatory and beneficial effect at the beginning of the disease.

### 2.4. Potential Support for Diagnosis Role of NLRP3 in Blood Samples from ALS Patients

Based on the findings obtained in the animal model, we aimed to translate this study to blood samples from ALS patients and to investigate the role of NLRP3 as a potential support for diagnosis biomarker in these samples.

We studied, by real-time PCR, the differences in *NLRP3* gene expression profiles between three different groups: healthy subjects (control), ALS patients and patients with other myopathies (OMP) (Table 1).

The results obtained showed a significant increase of *NLRP3* transcriptional levels in blood samples from ALS patients compared to healthy controls (Figure 4). Although a decrease was also observed in the OMP group, these differences did not reach a significant level. Furthermore, comparison of the *NLRP3* expression profile with the area under ROC curves (AUC) was performed to test if *NLRP3* gene levels could perform well as a support for diagnosis tools. AUC results were statistically significant with an area of 0.270 (95% confidence interval, CI: 0.081–0.459, *p* = 0.039).

Hence, these results suggested that *NLRP3* transcriptional levels in peripheral blood of ALS patients may be a potential support for diagnosis biomarkers.

## 3. Discussion

In recent years, it has been demonstrated that NLRP3 inflammasome plays an important role in diverse neurodegenerative diseases; however, its implication in ALS remains unclear. Thus, our first objective was to study the gene and protein expression of the NLRP3 inflammasome components in skeletal muscle from WT and SOD1G93A mice models throughout disease progression.

Upregulation of the inflammasome genes was observed at the asymptomatic stage (P50) of SOD1G93A mice (*Asc*, *caspase 1* and *Il1β* were significantly increased), so these genes may represent potential prognostic and diagnostic biomarkers of ALS (Figure 1A). However, mRNA abundance did not correlate with protein profile in this stage. Increased translational levels of inflammasome components were observed at advanced stages of the disease, which became significant at P120 (Figure 1B). It should be noted that mature IL1β was only detectable in SOD1G93A animals (Figure 1C), which is in accordance with previous studies performed in the skeletal muscle of SOD1G93A transgenic rats [14] and mice [15]. In the last study mentioned, they also analyzed the expression of different NLRs (NLRP1, NLRC4, NLRP3 and AIM2) and observed elevated levels of NLRC4, and to a lesser extent NLRP3, in gastrocnemius muscle. This difference with respect to our results in quadriceps could be explained by the fact that distinct types of skeletal muscle fibers may express diverse PRRs (including different TLRs and NLRs) in a function of environmental factors (such as pathogens, inflammatory cytokines and growth factors), so it is possible that a particular NLR/inflammasome predominates more than others depending on the muscle [16,17,18].

Therefore, the results obtained suggest that the progressive activation of NLRP3 inflammasome with the course of the disease could enhance muscle degeneration. This hypothesis is supported by various studies that have demonstrated that IL1β, after being activated by NLRP3 inflammasome, plays a crucial role in the onset and disease progression of several myopathies [19], such as myositis [20], sarcopenia [21] and Duchenne muscular dystrophy [22,23]. In fact, germline deletion of Nlrp3 in mice led to reduced IL1β serum levels and protects from inflammation-induced skeletal muscle atrophy [24]. Based on these findings, our hypothesis that the activation of the NLRP3 inflammasome would promote muscle degeneration in ALS is reinforced. Hence, NLRP3 may be pointed out as a possible therapeutic target in ALS skeletal muscle.

Regarding the discrepancy observed between the abundance of protein and mRNA, this fact is in accordance with several studies that have demonstrated that mRNA levels are not necessarily predictive for corresponding protein levels [25,26]. Furthermore, considering the alteration of the RNA metabolism that characterizes ALS pathogenesis, this result is not surprising. In fact, previous studies of inflammasome in ALS have not seen agreement between transcriptional and translational levels [9,15].

In view of these findings, our next step was to analyze the prognostic nature of the NLRP3 inflammasome component’s genes in serial biopsies of skeletal muscle from SOD1G93A mice. This methodology has already been translated to patients from the animal model for other candidate biomarkers [27,28].

*Nlrp3* transcriptional levels between the early symptomatic (75 days) and symptomatic (105 days) stages correlated significantly and positively with the longevity of transgenic mice (r = 0.506, *p* = 0.027) (Figure 2). This means that the animals that display higher transcriptional levels of this gene at earlier stages of the disease survive longer. On the one hand, this finding suggested that *Nlrp3* could be a potential prognostic biomarker of longevity in this animal model. On the other hand, contrary to what our hypothesis initially expected, NLRP3 would not be completely harmful and could exert a beneficial function for the muscle at the onset of the disease.

This may be explained by a recently proposed model that defends the notion that aging or excess nutrients constitute a tipping point to switch NLRP3 function from beneficial to pathological [29]. This model suggests that the primary physiological function of NLRP3 is to clear noxious substances (such as protein aggregates, a characteristic feature of ALS with SOD1G93A and TDP-43Q331K) and to regulate metabolism. However, with age or earlier in life if there are excess nutrients, NLRP3 becomes pathologic due to too strong of a response or to macrophage overactivation caused by the chronic presence of excessive amounts of these harmful substances. Therefore, it has been proposed that this NLRP3 beneficial function in homeostasis could become harmful during the onset of inflammatory and metabolic diseases, such as multiple sclerosis or Alzheimer’s disease. Interestingly, it has been seen that mutant proteins SOD1 and TDP-43 activate inflammasome in primary mouse microglia, accelerating ALS pathogenesis [30,31].

On the other hand, in a murine model of ALS in which neurodegeneration induces denervation and muscular atrophy, it is possible to think that the activation of the inflammasome may have a dual role, as has been described in ALS for its known and powerful activator, the P2X7 receptor. It has been found that the pharmacological antagonism of P2X7 exerts a positive result in the motor performance of SOD1G93A mice [32] and significantly prolongs the survival of females [33], while its constitutive deletion anticipates the disease onset and aggravates its progression [34]. Thus, it has been shown that P2X7 plays a double role in ALS pathogenesis, and its activation may be beneficial or harmful depending on the time.

This dual role of inflammasome is consistent with studies performed on other diseases such as cancer and Alzheimer’s disease, in which despite the fact that the inflammasome is directly involved in its pathogenesis, there are also reasons to speculate on whether a moderate degree of its activation could be beneficial, thus being NLRP3 inflammasome a double-edged sword [29,35,36,37,38]. In fact, several studies have shown that the activation of inflammasome, and therefore caspase 1, has unexpected consequences in certain cell types, since it not only induces inflammation by activation and secretion of proinflammatory cytokines, but also regulates levels of extracellular proteins involved in tissue repair and cytoprotection [29,39,40,41].

It is important to highlight that only *Nlrp3* significantly correlated with longevity, so it could exert a function, different and independent of inflammasome, that is beneficial for skeletal muscle. Recent studies have demonstrated that NLRP3 can be imported into the cell nucleus and act as a transcription factor in Th2 lymphocytes promoting the expression of IL-4 [42,43,44,45,46]. Interestingly, this anti-inflammatory cytokine is involved in muscle growth and regeneration [47,48,49] and, after spinal cord injury, it drives microglia and macrophages toward a favorable phenotype for tissue repair and functional recovery [50]. In fact, a recent study has revealed that IL-4 gene therapy in our mouse model delays the onset of ALS and ameliorates clinical outcomes during the early slowly progressive phase of the disease [51]. However, such modulation does not produce beneficial effects or reverse neurodegenerative processes in the late and fast progressing phase of the disease, which could explain why only the *Nlrp3* expression in skeletal muscle correlated with longevity in the early stages and not at the end.

In addition, it is noteworthy that a recent study has shown that stimulation of the P2X7 axis with an agonist in SOD1G93A mice significantly delays and prevents the denervation atrophy of skeletal muscles [52]. This finding could support our results about the positive correlation of *Nlrp3* levels and longevity in the earlier stages, since it has been observed that P2X7 activates NLRP3 through its interaction [53] or the flow of K+ which triggers after its opening [54]. This produces conformational changes in NLRP3, thus leaving its structure open and thereby enabling both its translocation into the nucleus and its oligomerization in the cytoplasm to give rise to the inflammasome assembly. However, other inflammasomes such as NLRP2, NLRC4, NLRP6 and AIM2 have been seen not to be activated by P2X7, with the possible exception of NLRP1 and NLRP2 [54]. In view of all this, it is possible that the muscle improvement observed in SOD1G93A mice after stimulation of P2X7 is closely related to NLRP3 activation.

For this reason, our next objective was to elucidate whether the activation of NLRP3 is beneficial during the disease progression in this murine model of ALS. To do this, we administered MCC950, a small molecule that specifically inhibits NLRP3 by closing its conformation and driving it toward an inactive state, to 70-day-old mice [12,13].

We observed that mice with inactivated NLRP3 lived less than those that had not received MCC950 treatment (Figure 3). It should be noted that this difference was greater especially in the earlier stages, which confirms the stated hypothesis that NLRP3 (either through the inflammasome or in an independent way) would exert a beneficial function in ALS pathogenesis particularly at the beginning of the disease. This result could support the idea that NLRP3 has a favorable function by acting as a transcription factor of IL-4, since it is in accordance with the previously mentioned gene therapy study with this cytokine performed on our animal model in which positive effects were observed mainly in the early slowly progressive phase of ALS [51].

On the other hand, it is possible that over time MCC950 has ceased to have effect on IL1β levels due to the activation of another NLR member/inflammasome as a compensatory response. In fact, it has been observed that in human blood with *S. typhimurium* treated with MCC950 there was a sharp decrease in IL1β at the beginning, but after a while their levels increased again due to the action of NLRC4 inflammasome [55]. This could explain the fact that in the last stages the difference between treated and untreated animals is somewhat smaller, since MCC950 might not be acting on the production of IL1β, and thus the disease pathogenesis would not be ameliorated under this treatment. This idea would be in accordance with a previous study where it was observed that SOD1G93A-mediated inflammasome activation (IL1β production) occurs independently of NLRP3, unlike ASC which was essential [31].

Therefore, the specific inhibition of NLRP3 does not seem to be a good therapeutic approach for ALS, and it would be better to act at other levels of the pro-inflammatory cascade of inflammasomes, such as caspase 1, IL1β [31] or all inflammasome components at once [56].

Based on all these findings, we decided to continue the study of NLRP3 as a promising biomarker for ALS patients in a less invasive tissue such as peripheral blood, which could help better the characterization of patients in clinical practice.

We found that *NLRP3* transcriptional levels were significantly increased in ALS patients’ blood with respect to controls, but not compared to other myopathies (Figure 4). Consequently, *NLRP3* may be pointed out as a potential biomarker to support the diagnosis of ALS patients.

This result is highly relevant since it is very difficult to discover sensitive and specific ALS biomarkers, which are currently presented as an essential tool for clinical trials and practice. Generally, the search and discovery of new biomarkers are hampered by the unknown etiopathogenesis of ALS, the great phenotypic variability between patients and the presence of common characteristics with other myopathies and neurodegenerative diseases.

Moreover, it is important to highlight the fact that *NLRP3* could be a blood-derived biomarker, since it has the advantage that it can be tested in a non-invasive way and the impact of tissues and systems that are harmed throughout disease progression, such as by neuroinflammation, muscle damage or microglial activation, can be monitored concurrently.

Nonetheless, more exhaustive studies with patient samples are required to validate the diagnostic nature of NLRP3 and even its potential as prognostic biomarker, as we have seen in the results from our ALS mouse model.

## 4. Materials and Methods

### 4.1. Animals

Transgenic mice B6SJL-Tg SOD1G93A were used in this study because they provide a suitable ALS disease model [57]. These mice express a high copy number of the G93A mutant form of human superoxide dismutase 1 (SOD1) and were purchased from The Jackson Laboratory (Bar Harbor, ME, USA). The colony was maintained by mating hemizygous SOD1G93A males with C57BL/6J × SJL/J F1 hybrid females (B6SJLF1) purchased from Janvier Labs (Janvier Labs, Le Genest-Saint-Isle, France). The offspring were identified by PCR amplification of DNA extracted from tail tissue as described in The Jackson Laboratory protocol. The mice were housed in a pathogen-free environment under a standard light/dark (12:12) cycle at the animal facilities in Centro de Investigación Biomédica de Aragón. Food and water were provided *ad libitum*.

All experimental procedures were approved by the Ethic Committee for Animal Experiments from the University of Zaragoza (6 February 2020) and were registered with the following code number (PI08/20). The care and use of animals were performed in accordance with the Spanish Policy for Animal Protection RD53/2013 and the EU Directive 2010/63.

### 4.2. Mouse Tissue Collection

#### 4.2.1. Extraction of Skeletal Muscle Samples in WT and Transgenic SOD1G93A Mice

Hemizygous SOD1G93A mice and age-matched non-transgenic WT control mice at the early symptomatic (P50), symptomatic (P75) and terminal (P120) stages (*n* = 4 transgenic SOD1G93A mice and 4 WT mice, balanced males and females, per stage) were used to study gene and protein expression throughout disease progression. For this, animals were euthanized with CO_2_ and all surgical material was sterilized before dissection. Then, the *Quadriceps femoris* was extracted, frozen in liquid nitrogen and stored at −80 °C. For tissue processing, each skeletal muscle sample was pulverized in liquid nitrogen using a Cellcrusher cryogenic tissue pulverizer (Cellcrusher, Cork, Ireland). Subsequently, the tissue was divided into two parts, one for the gene study and one for the protein study, thus reducing the possibility of variation as it was the same animal in the two studies.

#### 4.2.2. Extraction of Biopsies from Skeletal Muscle of SOD1G93A Mice

Twenty SOD1G93A mice were used to study the correlation of gene expression with longevity (*n* = 10 mice per sex). For this, we used a methodology validated in our laboratory; it has already been shown that the results obtained in this animal model are transferable to patients [27,28]. Animals were anesthetized with isoflurane and three biopsies from the *Gluteus superficialis* muscle were obtained per mouse, from a different hind limb each time, at three different ages that coincided with the early symptomatic stage (75 days), symptomatic stage (105 days) and terminal stage (endpoint age). The humane endpoint for these mice was defined as the moment that the animals were unable to right themselves within 30 s after being placed on their side and the age at this moment was considered as death for survival analysis [58]. We used the same protocol described in our previous study [27]. This procedure allowed the study of gene expression in the same animal, since the *Gluteus superficialis* muscle (unlike the *Quadriceps femoris*) is easy to reach and its manipulation does not prevent the mouse from moving properly after each surgery, so it was possible to keep the animal alive at different disease stages throughout the study.

### 4.3. Administration of MCC950

Twenty-seven animals were used in this experiment and were divided into two groups with balanced sexes: 14 SOD1G93A mice treated with the drug and 13 untreated SOD1G93A mice. Following the protocol described in a recent study [59], animals received intraperitoneal injections of sterile PBS (control) or MCC950 (10 mg/kg in PBS) three times per week from the age of 70 days until death (endpoint age).

### 4.4. Patients

A total of 42 individuals participated in this study, matched for age and sex whenever possible: 14 healthy controls, 14 patients with other myopathies (OMP) and 14 ALS patients (Table 1).

In all cases written informed consent was obtained according to Declaration of Helsinki principles and to Directive 2004/23/EC of the European Parliament and of the Council.

### 4.5. Blood Samples from Patients

We received blood samples frozen in Pax tubes; the PAXgene Blood RNA Kit (PreAnalytiX GmbH, 8634 Hombrechtikon, Switzerland) was used to extract the RNA fraction. In all samples, the cDNA was also obtained from 1 µg of total extracted RNA (High Capacity cDNA RT kit; Applied Biosystems, Madrid, Spain).

The analysis of blood samples from Centro Clinico NEMO (Milan, Italy) was approved by the Research Ethics Committee of the Community of Aragon (7 November 2018) and was registered with the following code number (PI18/078).

### 4.6. Gene Expression

For RNA extraction, muscle tissue was homogenized using a PRO200 homogenizer (PRO Scientific Inc, Oxford, USA). Regarding blood samples, the Blood RNA kit (PreAnalytiX GmbH, 8634 Hombrechtikon, Switzerland) was used starting from frozen Pax tubes. From here, the same protocol was used for samples of animals and patients.

RNA was extracted with TRIzol reagent (Invitrogen, Prat de Llobregat, Spain) and treated with Turbo DNA-free kit (Ambion, Madrid, Spain) to eliminate genomic DNA. Once purified, RNA was retrotranscripted using the Superscript First Strand kit (Invitrogen, Prat de Llobregat, Spain). Quantitative real-time PCR (qRT-PCR) was performed from 1:10 diluted cDNA in triplicates using StepOne Real-Time PCR (Life Technologies; Waltham, MA, USA). The TaqMan probes (Applied Biosystems, Madrid, Spain) used in this study are indicated in Table 2.

*Gapdh* and *β-actin* were used as housekeeping genes for normalization of animal data, and *GAPDH*, *HPRT1* and *TBP* were used to normalize patient data. The relative gene expression was determined using the 2-ΔΔCT method [60].

### 4.7. Protein Expression

For protein extraction, muscle powdered tissue was resuspended in RIPA lysis buffer containing protease inhibitors (Roche, Basel, Switzerland), homogenated and centrifuged at 10,000× *g* for 10 min at 4 °C. The supernatant was collected and the protein concentration was determined by BCA method (Sigma-Aldrich, San Luis, MO, USA). After quantification, 25 μg of protein were loaded to an 8–10% SDS-PAGE and transferred to PVDF membranes (Amersham™, GE Healthcare Life Sciences, Little Chalfont, UK). For immunodetection, membranes were blocked with a Tris-buffered saline solution containing 5% skimmed milk and 0.1% Tween for 1 h at room temperature and then incubated overnight at 4 °C with the selected primary antibodies (Table 3). GAPDH was selected as the normalization protein. The next day, membranes were washed and incubated with HRP-conjugated anti-rabbit secondary antibody (sc-2004; Santa Cruz Biotechnology, Quimigen S.L., Madrid, Spain) for 1 h at room temperature, and the protein bands were visualized using ECL reagents (GE Healthcare Life Sciences, Little Chalfont, United Kingdom). Densitometry study was performed using AlphaEase FC software (Bonsai Technologies Group, Madrid, Spain).

### 4.8. Statistical Analysis

Comparisons of results obtained in expression studies between the WT and SOD1G93A groups or between healthy, ALS and other-myopathies patients were made using Student’s *t*-test, and Dunnet’s post hoc test was used for statistical correction. The area under ROC curves (AUC) was calculated to explore the support for diagnosis nature of NLRP3 in blood samples from healthy, ALS and other-myopathies patients. The Spearman correlation coefficient was used to determine the linear correlation between transcriptional expression levels and longevity of SOD1G93A mice. Survival over time was computed using Kaplan–Meier estimates, and the survival distributions of treated vs. non-treated animals were tested with the Mantel-Cox log-rank test. The software used for the statistical analysis was SPSS 22.0 (IBM, Barcelona, Spain). All data represent the means ± standard error of the mean (SEM). Values were considered statistically significant (*) at *p* < 0.05 and statistically highly significant at *p* < 0.01.

## 5. Conclusions

In view of the results obtained in the present study, it can be concluded that NLRP3 may be involved in the skeletal muscle pathogenesis of ALS, either through inflammasome or independently. Inactivation of NLRP3 could not be beneficial to ameliorate disease progression, but its gene expression might represent a potential biomarker for diagnosis and prognosis of longevity in skeletal muscle from animal models.

Although further research is needed, this study reveals for the first time that *NLRP3* could be useful for diagnostic support in blood samples from ALS patients, which may be of help in clinical trials and practice.

## Figures and Tables

**Figure 1 ijms-22-02523-f001:**
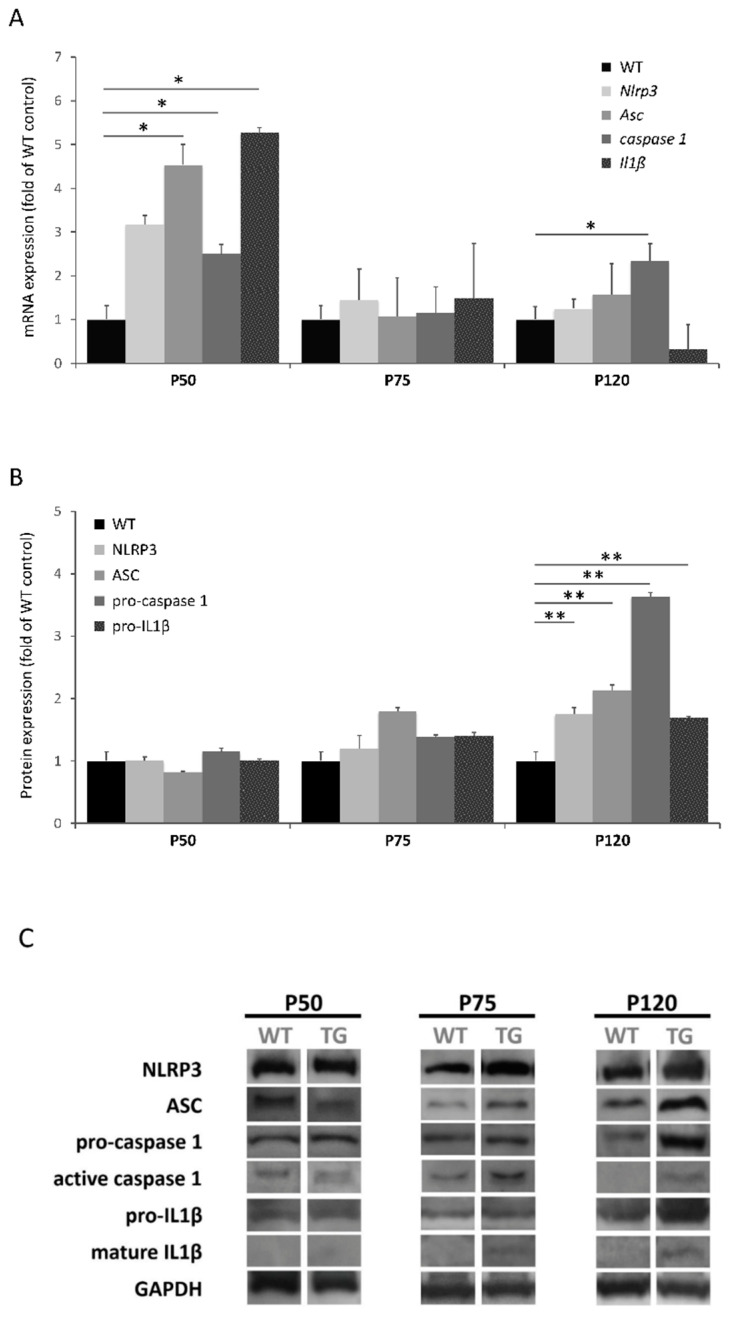
Expression levels of NLRP3 inflammasome in skeletal muscle of SOD1G93A mice during disease progression (P50: asymptomatic stage; P75: early symptomatic stage; P120: terminal stage). (**A**) Transcriptional levels of NLRP3 inflammasome components determined by real-time PCR in the quadriceps tissue. Dunnet’s post hoc test correction between groups also yielded statistically significant differences among groups at the asymptomatic stage (*Asc*, *p* = 0.037; *caspase 1*, *p* = 0.046; *Il1β*, *p* = 0.041). (**B**) Protein expression levels of NLRP3 components in quadriceps muscle determined by Western blotting. Dunnet’s post hoc test correction between groups also yielded statistically significant differences among groups at the terminal stage (NLRP3, *p* = 0.025; ASC, *p* = 0.007; pro-caspase 1, *p* = 0.003; pro-IL1β, *p* < 0.000). (**C**) Protein bands of each NLRP3 inflammasome component. Active caspase 1 and mature IL1β were only detected in SOD1G93A mice. The total number of animals used was 24 (8 mice for each stage: WT *n* = 4 and SOD1G93A *n* = 4). * *p* < 0.05, ** *p* < 0.01; significant differences in comparison to WT mice.

**Figure 2 ijms-22-02523-f002:**
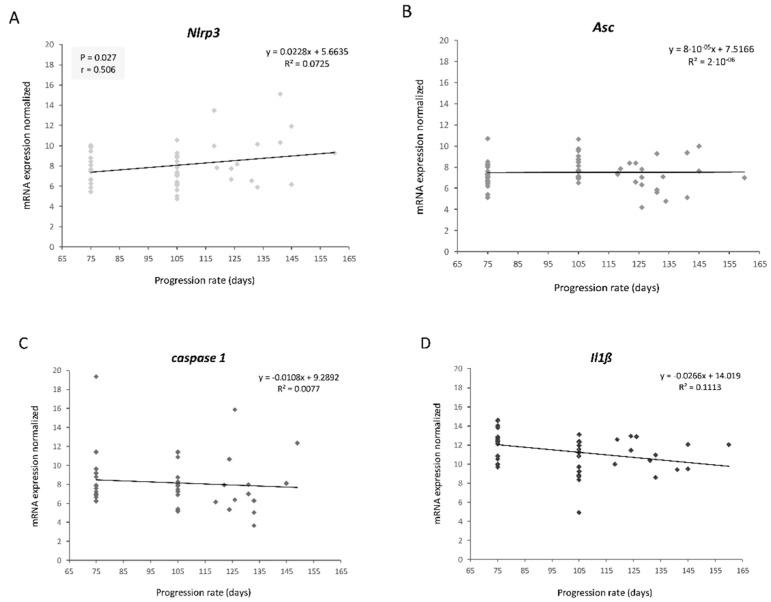
Linear correlation graphs between longevity and transcriptional expression levels of NLRP3 inflammasome components throughout disease progression: (**A**) *Nlrp3*, (**B**) *Asc*, (**C**) *caspase 1*, (**D**) *Il1β*. Only *Nlrp3* levels showed a significant and positive correlation with longevity. The total number of transgenic SOD1G93A mice used was 20. Three skeletal muscle biopsies were obtained per mouse, at different stages: early symptomatic stage (75 days), symptomatic stage (105 days) and terminal stage (endpoint age).

**Figure 3 ijms-22-02523-f003:**
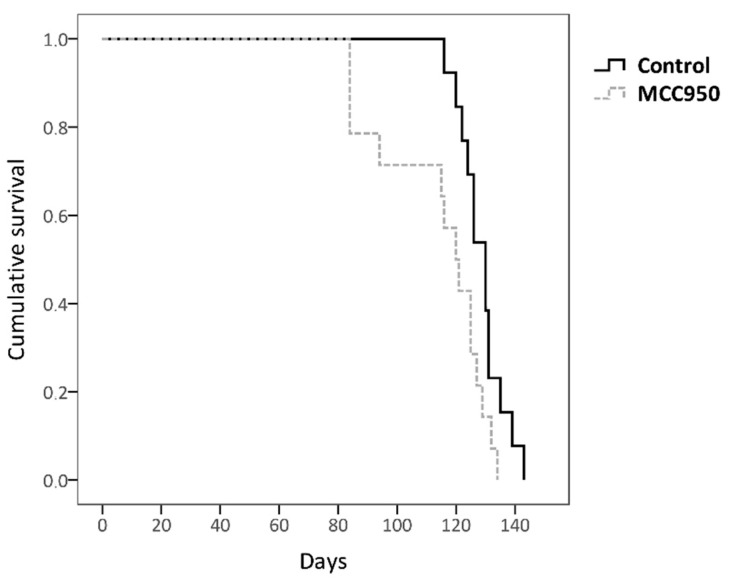
Kaplan–Meier analysis of the effect of MCC950 drug on survival rate. Non-treated transgenic SOD1G93A mice showed a longer lifespan (128.692 days ± 7.54) than MCC950-treated mice (113.571 days ± 18.48); log-rank test *p* = 0.025. The total number of animals used was 27 (control *n* = 13 and MCC950 treatment *n* = 14).

**Figure 4 ijms-22-02523-f004:**
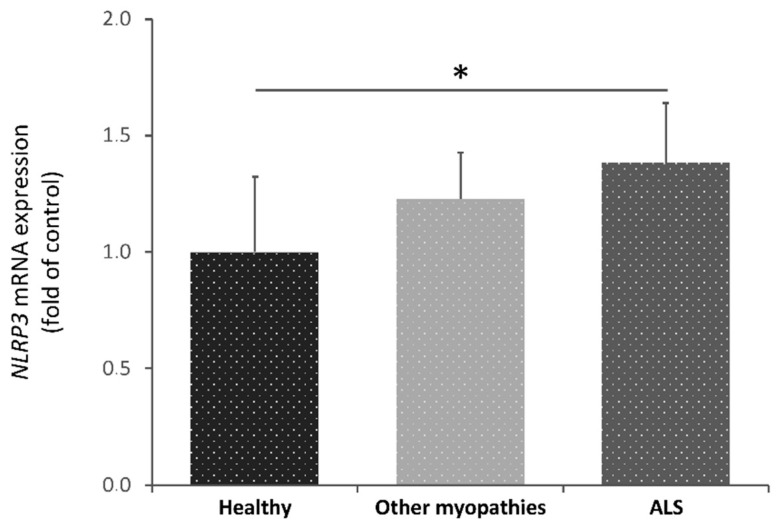
Transcriptional expression levels of *NLRP3* in blood samples from healthy subjects (control), amyotrophic lateral sclerosis (ALS) patients and patients with other myopathies. Kruskal-Wallis tests showed significant differences among ALS patients and the healthy group (*p* = 0.03). Dunnet’s post hoc test correction between groups also yielded statistically significant differences between these groups (*p* = 0.04). A total of 42 participants were included in this study: 14 control individuals, 14 patients with other myopathies and 14 ALS patients * *p* < 0.05.

**Table 1 ijms-22-02523-t001:** General and Clinical Characteristics of the Participants in the Blood Study Cohort.

Patients’ Characteristics	ALS Patients (*n* = 14)	OMP Patients (*n* = 14)	Control Subjects (*n* = 14)
Gender (*n*)	9 males	10 males	6 males
	5 females	4 females	8 females
Age at illness onset (mean ± SD)	62 ± 8.26		
Disease duration, months (mean ± SD)	27 ± 19.18		
Age at sampling (mean ± SD)	65 ± 8.23	57 ± 13.61	58 ± 9.53
Site at illness onset (*n* patients) Bulbar Upper limb Lower limb Generalized			
	2		
	4		
	7		
	1		

**Table 2 ijms-22-02523-t002:** Taqman^®^ Probe and Primer Mixtures Used in Gene Expression Assays.

Gene	Symbol	Organism	Assay ID
*NLR family, pyrin domain containing 3*	*Nlrp3*	*Mus musculus*	Mm00840904_m1
*PYD and CARD domain containing*	*Pycard*	*Mus musculus*	Mm00445747_g1
*caspase 1*	*Casp1*	*Mus musculus*	Mm00438023_m1
*Interleukin 1 beta*	*Il1β*	*Mus musculus*	Mm00434228_m1
*Glyceraldehyde-3-phosphate dehydrogenase*	*Gapdh*	*Mus musculus*	4352932E
*Actin, beta, cytoplasmic*	*ß-actin*	*Mus musculus*	4352933E
*NLR family pyrin domain containing 3*	*NLRP3*	*Homo sapiens*	Hs00918082_m1
*Glyceraldehyde-3-phosphate dehydrogenase*	*GAPDH*	*Homo sapiens*	Hs02786624_g1
*Hypoxanthine phosphoribosyltransferase 1*	*HPRT1*	*Homo sapiens*	Hs02800695_m1
*TATA-box binding protein*	*TBP*	*Homo sapiens*	Hs00427620_m1

**Table 3 ijms-22-02523-t003:** Primary Antibodies Used in Protein Expression Assays (Santa Cruz Biotechnology, Quimigen S.L., Madrid, Spain).

Antibody	Host	Dilution	Reference Number
Cryopyrin (H-66)	Rabbit	1:250	sc-66846
ASC (N-15)-R	Rabbit	1:250	sc-22514-R
caspase-1 p20 (M-19)	Rabbit	1:250	sc-1218-R
IL-1ß (H-153)	Rabbit	1:250	sc-7884
GAPDH (FL-335)	Rabbit	1:1000	sc-25778

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
