# Peer review of "Inflammasome in ALS Skeletal Muscle: NLRP3 as a Potential Biomarker"

_ijms, 2021, doi:10.3390/ijms22052523_

Round 1

Reviewer 1 Report

The authors claim that NLRP3 transcript expression levels in circulation and skeletal muscle can used as potential diagnostic biomarker to identify the early disease progression and prognosis of amyotrophic lateral sclerosis (ALS).  Authors concluded their hypothesis supporting based on the study in animal model and transcript analysis of blood samples from ALS patients. After extensive review of this current draft, the resulted data shown is interesting and convincing to support their hypothesis. The current manuscript written well and organized together a nice approach. Therefore, this revised draft accepted as publication without any major or minor comments, in to peer review journal Molecular Sciences from MDPI. 

Author Response

We are very grateful for the effort and time spent by the Reviewer to revise this manuscript and for the kind comments addressed.

Reviewer 2 Report

This is an excellent comprehensive study tackling inflammasone in skeletal muscle.  The authors comprehensive approach of assessing, IL1-beta, caspase, NLRP, and more in single study increases overall impact.  The authors also produce a summary analysis examining NLRP in 42 clinical patients.  The study has an overall excellent presentation and clarity.  

However, there are some statistical issues that need addressed and clarified in the methods.  

First, the authors do not state if they corrected their pairwise comparison p-value threshold for significance for multiple comparisons.  With multiple comparisons, there is an increased chance for type I error (false positive).  The authors should correct student's t-tests and Krusakl-Wallis p-value thresholds with either a Boneferroni correction, Tukey's post-hoc or a similar standardized statistical correction for multiple comparisons.   This appears to be needed for Figure 1A, 1B and Figure 4.  Correction factors will change the significance for some of these values that are very close to 0.05.  

The p<0.1 denoted as a "tendency" is completely misleading, especially if values have not been corrected for multiple comparison.   If the authors insist on reporting at an alpha value of 0.1 (p<0.1), simply state the two alpha levels (for example 0.05 and 0.1, prior to multiple comparison correction) and throw out the "tendency" terminology which has no standardized statistical meaning.

Though I'll leave it as "optional" to perform, a principal component analysis for the data shown in Figure 2 would probably tell more insight about how each of the metrics contributes to the variance of longevity.

Finally, in the Discussion, the authors may wish to compare their IL1-Beta results to the meta-analysis done for ALS mouse models by Jeyachandran 2015, Frontiers in Cellular Neuroscience 9:42:  https://doi.org/10.3389/fncel.2015.00462

Author Response

Response to Reviewer 2 Comments

Point 1: The authors do not state if they corrected their pairwise comparison p-value threshold for significance for multiple comparisons.  With multiple comparisons, there is an increased chance for type I error (false positive).  The authors should correct student's t-tests and Krusakl-Wallis p-value thresholds with either a Boneferroni correction, Tukey's post-hoc or a similar standardized statistical correction for multiple comparisons.   This appears to be needed for Figure 1A, 1B and Figure 4.  Correction factors will change the significance for some of these values that are very close to 0.05.

Response 1: We fully agree with this comment. Therefore, we have performed correct student's t-tests and Krusakl-Wallis p-value thresholds with Dunnet’s post hoc test correction for multiple comparisons and we have obtained the same results. The reason of using Dunnet’s post hoc test is based on the fact that we have compared transgenic mice versus control mice or ALS patients versus healthy controls. For this reason, the most suitable test seems to be Dunnet’s test.

Point 2: The p<0.1 denoted as a "tendency" is completely misleading, especially if values have not been corrected for multiple comparison. If the authors insist on reporting at an alpha value of 0.1 (p<0.1), simply state the two alpha levels (for example 0.05 and 0.1, prior to multiple comparison correction) and throw out the "tendency" terminology which has no standardized statistical meaning.

Response 2: The reviewer is right and we agree that the terminology "tendency" does not have a standardized statistical meaning. Therefore, we have eliminated the terminology "tendency" in the manuscript.

Point 3: Though I'll leave it as "optional" to perform, a principal component analysis for the data shown in Figure 2 would probably tell more insight about how each of the metrics contributes to the variance of longevity.

Response 3:  We are very grateful for the Reviewer’s comment. We contemplated this analysis at the beginning of this study although it was not possible to perform it because of the number of samples and number of biomarkers selected. As far as we know, this analysis provides you a dimensionality-reduction method that is often used to reduce the dimensionality of large data sets. We hope to use it in future studies in human samples by using a large number of samples, even serial samples.

Point 4: In the Discussion, the authors may wish to compare their IL1-Beta results to the meta-analysis done for ALS mouse models by Jeyachandran 2015, Frontiers in Cellular Neuroscience 9:42:  https://doi.org/10.3389/fncel.2015.00462

Response 4: Thank you very much for the suggestion, the article is really very interesting. However, we think that it does not quite fit well with IL-1β gene and protein expression levels obtained in our article. Nevertheless, if you really consider that it would be advisable to include it, we will find a way to do it.

Reviewer 3 Report

In this manuscript, Moreno-Garcia et al. proposed using NLRP3 as a potential biomarker for ALS diagnosis and prognosis due to its role in several neurodegenerative diseases. To support their hypothesis, they measured the gene and protein expression of NLPR3 inflammasome components in muscle tissues from G93A-SOD1 mice, as well as the serum NLRP3 mRNA level in patients. They further inactivated NLRP3 with MCC950, a specific conformation-dependent inhibitor, in G93A mice, and found a correlation with a shorter life span in these mice, particularly at the early stage. Finally, they concluded that NLRP3 could play a dual role during disease progression and its gene expression level could be used as a biomarker. While the need to find an ALS biomarker for diagnosis and prognosis is high, thus making this manuscript interesting. The conclusion is not well supported due to the insufficient data points, lack of male/female comparison, limited sample size, mixed sample types, small differences between groups, and the complicated nature of NLRP3. Furthermore, serum NLRP3 mRNA level is unlikely to be a clinical biomarker for ALS, due to the dual role of NLRP3 at various stages, its non-specific role in ALS, and the small difference in the serum mRNA level between ctrl and ALS patients as well as large individual variability within the same group. Therefore, the manuscript needs to be revised based on several major points listed below:

  1. Mixed genders. As inflammation and immune responses are heavily gender-dependent. Male and female mice should be analyzed separately with increased Ns. Gender differences should also be taken into account in the human serum NLRP3 mRNA study.
  2. Muscle vs. serum. While it is not surprising that NLRP inflammasome is activated in muscle from ALS mice, it is less clear whether the serum level would be a good indicator. Given the small (though significant) difference between human control and ALS groups, it is worth investigating whether the serum NLRP3 mRNA level correlates with disease progression and the muscle NLRP3 level in a well-characterized model (e.g. the G93A-SOD1 model here).  
  3. mRNA vs. protein. It is interesting that the authors found dramatic differences in inflammasome components at the mRNA level between control and asymptomatic G93A mice and that difference disappeared at the early symptomatic stage and became much smaller at the late stage (the difference in NLRP3 is not significant between control and G93A). In contrast, the protein level seems to differ more at the late stage and perhaps also during the early stage (note that the error bars were also smaller, as compared with the mRNA data). Why did the authors choose mRNA for patient serum instead of protein, as most of those subjects are symptomatic already? Also, why not use mature IL1β or caspase 1?  Furthermore, muscle nlrp3 level did show some correlation with longevity, but it is rather weak, plus the patient serum NLRP3 mRNA data showed only small differences between control and ALS groups with big error bars, all these make NLRP3 a less promising biomarker. 
  4. Time course. Given the dual role of NLRP3, it is puzzling that the authors picked only one time point (70day, early symptomatic stage) to inhibit NLRP3 protein activity pharmacologically. A more complete dataset including presymptomatic and late stages, a dose-dependent curve, and treatment in WT mice is needed to determine the complex role of NLRP3 in ALS.
  5. ALS types. As most of the ALS patients are sporadic, all the animal data showed here were restricted to the G93A-SOD1 model, it would be more informative if other SOD1 animal models or human muscle-MN organoids from both familial and sporadic cases are also used. 
  6. ALS vs. other myopathies and neurodegenerative diseases. The current study as well as pre-existing literature have all suggested that NLRP3 inflammasome is not specific for ALS. A good biomarker for disease diagnosis and prognosis needs to have good specificity and selectivity. NLRP3 seems to have neither. 

In conclusion, though the topic is interesting and the data have a certain level of merit, the conclusion that NLRP3 could be an ALS biomarker is not sound. The authors need to design the experiments more thoroughly, analyze their data more vigorously, discuss the results more critically, and draw conclusions more carefully.

Author Response

Response to Reviewer 3 Comments

Point 1: Mixed genders. As inflammation and immune responses are heavily gender-dependent. Male and female mice should be analyzed separately with increased Ns. Gender differences should also be taken into account in the human serum NLRP3 mRNA study.

Response 1: We thank the reviewer for this comment. In this study, the number of samples collected was finally not large enough to analyze the expression profile of all the biomarkers suggested in male and female separately. In the preclinical study with transgenic mice, the ethics committee allowed us to perform the study with the number of animals described on it. The new animal welfare rules followed in our Institution promote preclinical studies that enroll a minimum number of mice. Therefore, we could not increase the number of animals and consequently the number of samples. Regarding the preliminar study with ALS patients, all the samples included in the study were the samples we have collected till now. We expect to reclute more samples in future studies, albeit in the case of ALS which is a rare neurodegenerative disease, it is difficult to obtain a large number of samples from ALS patients unless a multicenter study is performed.

Point 2: Muscle vs. serum. While it is not surprising that NLRP inflammasome is activated in muscle from ALS mice, it is less clear whether the serum level would be a good indicator. Given the small (though significant) difference between human control and ALS groups, it is worth investigating whether the serum NLRP3 mRNA level correlates with disease progression and the muscle NLRP3 level in a well-characterized model (e.g. the G93A-SOD1 model here). 

Response 2: We agree that it would be interesting to investigate whether the serum NLRP3 mRNA level correlates with disease progression in the SOD1G93A model. However, we first aimed to study a more affected tissue by disease progression such as the skeletal muscle and therefore, we performed serial muscle biopsies. In addition, previous studies in our group enabled the translational study from the animal model to ALS patients by following the same methodology presented in this study (as reflected in the article "Collagen XIX Alpha 1 Improves Prognosis in Amyotrophic Lateral Sclerosis" doi: 10.14336/AD.2018.0917). Therefore, we consider appropriate to carry out this study in the same way.

Point 3: mRNA vs. protein. It is interesting that the authors found dramatic differences in inflammasome components at the mRNA level between control and asymptomatic G93A mice and that difference disappeared at the early symptomatic stage and became much smaller at the late stage (the difference in NLRP3 is not significant between control and G93A). In contrast, the protein level seems to differ more at the late stage and perhaps also during the early stage (note that the error bars were also smaller, as compared with the mRNA data). Why did the authors choose mRNA for patient serum instead of protein, as most of those subjects are symptomatic already? Also, why not use mature IL1β or caspase 1?  Furthermore, muscle nlrp3 level did show some correlation with longevity, but it is rather weak, plus the patient serum NLRP3 mRNA data showed only small differences between control and ALS groups with big error bars, all these make NLRP3 a less promising biomarker.

Response 3: We agree with the Reviewer with this comment and we would like very much to analyze protein levels in blood samples from patients. Notwithstanding, the blood samples were collected in Pax tubes and these tubes are quite suitable for preserving RNA albeit it is uncertain if they are suitable for preserving protein levels. In fact, we tried to extract proteins from these samples but the concentration was extremely low for the majority of the samples or non-existant in some cases. We are afraid this protein study could not be performed in these samples. This is the reason why we focused the study at gene expression level.

Regarding the analysis of mature IL1b or caspase 1, we studied NLRP3 in samples from patients because NLRP3 gene levels showed a significant correlation with the longevity in mice that were in accordance with the preclinical study based on NLRP3 inactivation. In other words, among the four components of inflammasome, only NLRP3 transcriptional levels correlated with longevity and then this effect on longevity was in line with NLRP3 inactivation by MCC950. On the other hand, and considering that we could only study gene expression in samples from patients since they were conserved in PAX tubes, it is not possible to analyze mature caspase 1 and IL1β at gene expession level. We are aware of the weak correlation of NLRP3 gene levels with the longevity of the animal model and of the small differences between control and ALS group that also show big errors bars quite probably due to the variability of the samples. We believe that albeit the results are not high significant, in the case of ALS disease that has no known origin, any promising result needs to be investigated in future studies with large number of samples. From our humble point of view, small although significant results in a reduced group of samples can open the door to promising molecular candidates in this disease, such as NLRP3.

Point 4: Time course. Given the dual role of NLRP3, it is puzzling that the authors picked only one time point (70day, early symptomatic stage) to inhibit NLRP3 protein activity pharmacologically. A more complete dataset including presymptomatic and late stages, a dose-dependent curve, and treatment in WT mice is needed to determine the complex role of NLRP3 in ALS.

Response 4: We agree with the Reviewer in this comment. From our expertise in this animal model, we usually start potential treatments in the symptomatic stage (60-70 postnatal days) because if this treatment is successful this treatment is easier to translate at clinical level at which ALS patients are still symptomatic. In this sense, we select the most appropriate age in the animal model that can be representative for the ALS patient, once diagnosed. For this reason, the age of starting the treatment is the symptomatic stage. In relation to the dose-dependent curve of the NLRP3 inhibitor, we did not perform this curve because we followed a previous methodology tested in another animal model to validate it in our model of ALS.

Point 5: ALS types. As most of the ALS patients are sporadic, all the animal data showed here were restricted to the G93A-SOD1 model, it would be more informative if other SOD1 animal models or human muscle-MN organoids from both familial and sporadic cases are also used.

Response 5: We agree with this comment. In fact, in the future if we obtain samples from sporadic patients, we will undoubtedly do a more exhaustive study of NLRP3. Although the results showed here are only from genetic cases, they are very encouraging since our research group has experience in transferring the results obtained in familiar cases to sporadic patients, as reflected in the article "Collagen XIX Alpha 1 Improves Prognosis in Amyotrophic Lateral Sclerosis" (doi: 10.14336 / AD.2018.0917), albeit our animal model suffers from the disease due to a genetic cause.

Point 6: ALS vs. other myopathies and neurodegenerative diseases. The current study as well as pre-existing literature have all suggested that NLRP3 inflammasome is not specific for ALS. A good biomarker for disease diagnosis and prognosis needs to have good specificity and selectivity. NLRP3 seems to have neither.

Response 6: Although it is true that NLRP3 has been studied in other diseases, we consider that the results obtained here are relevant in the field of ALS. We have proposed NLRP3 as a promising prognostic biomarker of longevity in the mice and as a support for diagnosis in the clinical practice, but we are aware future studies are needed to validate this biomarker as a good biomarker for this disease. Till date, the only accepted biomarker of ALS is the neurofilamment Light chain. However, this biomarker is not suitable for all the affected tissues and in this regard, the identification of more potential biomarkers of this disease is needed. On the other hand, ALS disease can show similar molecular targets with other neurodegenerative and neuromuscular diseases since the molecular mechanisms altered in ALS are common to other diseases with similar ethiology. Consequently, it can be really difficult to identify a very specific biomarker of ALS rather than a combination of biomarkers that can help for the diagnosis and or prognosis of the disease. In addition, and in accordance with a comment of another Reviewer that has also revised this manuscript kindly, we have performed a ROC curve to further validate the potency of NLRP3 as support for diagnosis biomarker in ALS.

Finally, as the Reviewer suggested, we have revised the data, as well as the discussion and conclusion sections to better adapt to the findings obtained in this study.

Round 2

Reviewer 3 Report

The revised manuscript hasn't added the data needed to sufficiently support their hypothesis. However, the reviewer understands the challenges raised by the authors and appreciates the changes in the text to better reflect the data. While the reviewer is still not convinced that NLRP3 gene expression is a good biomarker for ALS, it is up to the scientific community to judge. Therefore, an argument can be made to publish this manuscript as it is so that other groups can either try to reproduce the data or examine changes in inflammasome components in their preexisting datasets obtained from larger cohorts of ALS mice and patient samples to compare with the results found here.